# Can Lung Ultrasound Be the Ideal Monitoring Tool to Predict the Clinical Outcome of Mechanically Ventilated COVID-19 Patients? An Observational Study

**DOI:** 10.3390/healthcare10030568

**Published:** 2022-03-18

**Authors:** Luigi Vetrugno, Francesco Meroi, Daniele Orso, Natascia D’Andrea, Matteo Marin, Gianmaria Cammarota, Lisa Mattuzzi, Silvia Delrio, Davide Furlan, Jonathan Foschiani, Francesca Valent, Tiziana Bove

**Affiliations:** 1Dipartimento di Scienze, Orali e Biotecnologiche, Università degli Studi “G. d’Annunzio”, 66100 Chieti, Italy; luigi.vetrugno@unich.it; 2Anesthesia and Intensive Care Clinic, Department of Medicine, University of Udine, 33100 Udine, Italy; sd7782.do@gmail.com (D.O.); natasciadandrea@yahoo.it (N.D.); matteo.marin.vr@gmail.com (M.M.); mattuzzi.lisa@gmail.com (L.M.); silviadelriomd@gmail.com (S.D.); furlan.dav@gmail.com (D.F.); jonathan.foschiani@gmail.com (J.F.); tiziana.bove@uniud.it (T.B.); 3Division of Anesthesia, Analgesia and Intensive Care, Department of Medicine and Surgery, University of Perugia, 06123 Perugia, Italy; gianmaria.cammarota@unipg.it; 4Clinical and Evaluational Epidemiologic Service, Department of Governance, Local Health Authority, 38123 Trento, Italy; francesca.valent@apss.tn.it

**Keywords:** COVID-19, critical care, lung ultrasound, lung ultrasound score, acute respiratory distress syndrome

## Abstract

Background: During the COVID-19 pandemic, lung ultrasound (LUS) has been widely used since it can be performed at the patient’s bedside, does not produce ionizing radiation, and is sufficiently accurate. The LUS score allows for quantifying lung involvement; however, its clinical prognostic role is still controversial. Methods: A retrospective observational study on 103 COVID-19 patients with respiratory failure that were assessed with an LUS score at intensive care unit (ICU) admission and discharge in a tertiary university COVID-19 referral center. Results: The deceased patients had a higher LUS score at admission than the survivors (25.7 vs. 23.5; *p*-value = 0.02; cut-off value of 25; Odds Ratio (OR) 1.1; Interquartile Range (IQR) 1.0−1.2). The predictive regression model shows that the value of LUSt0 (OR 1.1; IQR 1.0–1.3), age (OR 1.1; IQR 1.0−1.2), sex (OR 0.7; IQR 0.2−3.6), and days in spontaneous breathing (OR 0.2; IQR 0.1–0.5) predict the risk of death for COVID-19 patients (Area under the Curve (AUC) 0.92). Furthermore, the surviving patients showed a significantly lower difference between LUS scores at admission and discharge (mean difference of 1.75, *p*-value = 0.03). Conclusion: Upon entry into the ICU, the LUS score may play a prognostic role in COVID-19 patients with ARDS. Furthermore, employing the LUS score as a monitoring tool allows for evaluating the patients with a higher probability of survival.

## 1. Introduction

The COVID-19 pandemic increased the workload of intensive care units (ICU) with a mean admittance of 16% of SARS-CoV-2-positive hospitalized patients. The principal diagnosis at admission was respiratory insufficiency [1]. The pragmatic reference standard for diagnosing the infection is the nasopharyngeal molecular swab test, while the chest computed tomography (CT) scan is the gold standard for diagnosing COVID-19 pneumonia [2]. Lung ultrasound (LUS) is a well-established diagnostic tool in acute respiratory failure and acute respiratory distress syndrome (ARDS), suited for COVID-19 clinical management giving results similar or superior to chest CT and superior to traditional chest X-rays [3,4]. LUS is a useful tool: it can be performed at the bedside, is easy-to-learn, is easy-to-use, is radiation-free, gives relevant clinical information, and permits saving precious time [5].

The LUS score allows for examining and rating the pulmonary aeration. It is calculated by dividing the thorax into 12 regions and assigning a number from 0 (normal lung) to 3 (lung consolidation) to each region; the sum of these numbers gives a value that ranges from 0 (completely aerated lung) to 36 (completely consolidated lung) [6].

During the pandemic, specific sonographic patterns were described [7]. However, the application of LUS in COVID-19 patients is still controversial. Persona et al. suggested that the LUS score is not as reliable as in the non-COVID-19 patients [8], whereas Stecher et al. stated that the LUS in ICU COVID-19 patients predicts the clinical course but not the outcome [9]. 

However, in a study conducted in Israel in a medical ward and intensive care setting, the baseline LUS score predicted clinical deterioration and death [10].

This study aimed to evaluate whether the LUS score at the ICU admittance can predict the clinical outcome in COVID-19 patients. The secondary aim was to evaluate a correlation between LUS score trends and clinical course in the survived patients.

## 2. Materials and Methods

### 2.1. Study Protocol 

This study was a retrospective observational study of prospectively and systematically collected data about LUS examination in patients with SARS-CoV-2 admitted to the Department of Anesthesia and Intensive Care of the University Hospital of Udine, Italy. The Institutional Review Board of the University of Udine approved the study with the number ID # 068/2021, 8 September 2021. The patient’s consent was obtained through the general consent (GECO) system, and the European General Data Protection Regulation 2016/679 (GDPR) was respected.

### 2.2. Study Population 

Inclusion criteria were: patients admitted to the COVID-19 ICU with a positive nasopharyngeal molecular swab test, patients with > 18 years of age. 

Exclusion criteria were: history of lung surgery (lung resections or pneumonectomy), severe pulmonary fibrosis and lung cancer or metastatic localization, difficult ultrasonographic window.

### 2.3. Lung Ultrasound Examination 

Experienced intensive care physicians performed LUS with an Affiniti 70 G ultrasound machine (Philips, Amsterdam, The Netherlands) with a convex probe (Mhz 2−5).

As a normal clinical practice, we calculate the LUS score on the day of admittance and discharge from the ICU. For the patients who died, we reported only the first LUS score evaluation. 

Before starting the enrollment, we organized a discussion between the operators about the LUS approach. To test the LUS inter-operator variability in the interpretation of LUS signs and patterns, online training was set up with a total of 25 clips, including the whole range of significant LUS COVID-19 signs.

We calculated the LUS score by dividing the thorax into 12 regions, 6 for each hemithorax, through the anterior and posterior axillary lines and a transverse line starting from the xiphoid process.

That results in three superior areas (anterior, lateral, and posterior) and three inferior areas (anterior, lateral, and posterior) for each hemithorax, permitting a global evaluation. 

The international evidence-based recommendations for point of care lung ultrasound [11] describe the possible ultrasound patterns and profiles that may be found, attributing a score from 0 to 3: 0 points for a normal or A-pattern (A-lines or <2 B lines and lung sliding present), 1 point for B1-pattern (well-spaced ≥ 3 B lines and lung sliding present), 2 points for B2-pattern (coalescent B-lines, lung sliding present, and light beam), 3 points for C-pattern (lung consolidation and multiple subpleural consolidations). 

Adding the different scores, we obtained the LUS score ranging from 0 (normal lung) to 36 (completely consolidated lung). 

### 2.4. Recorded Data

Anthropometric parameters such as age, gender, weight, height, and body mass index (BMI) were recorded, and medical history and clinical conditions at the admission. 

We reported the necessity of oxygen therapy, non-invasive ventilation (NIV), intubation and mechanical ventilation, and the specific therapy duration. 

C-reactive protein (PCR), D-dimer, and interleukin 6 (IL-6) at admission and discharge were registered. 

### 2.5. Study Outcome 

The main aim was to verify if the LUS score at admission in the ICU could predict the clinical outcome (survival or death) in critically ill COVID-19 patients. The secondary aim was to evaluate the trend of LUS score at admission and at discharge from the ICU of the survived patients to verify if there was a correlation between the LUS score and the disease course.

### 2.6. Statistical Analysis 

The distribution between the two groups of patients (intra-ICU survivors and deceased) was compared by Student’s *t*-test (variables are expressed as mean and standard deviation) after verifying the normality of the distribution by means of the Shapiro–Wilk test. In contrast, Fisher’s exact test was used for variables expressed as absolute frequency and relative percentage, and a *p*-value < 0.05 was considered statistically significant. We also verified the correlation between the measured variables and the outcome by univariate and multivariate logistic regression.

The paired Student’s *t*-test was used to compare the LUS at admission (t0) and the discharge from the ICU (t1).

All statistical analyses were performed using open-source software “R: A language and environment for statistical computing”, implementing the “readODS”, “compareGroups”, “ggplot2”, “ggExtra” packages.

## 3. Results

From 1 December 2020 to 30 April 2021, 104 patients were enrolled in the study. One patient was excluded from the analysis due to incomplete clinical data, and therefore the final sample consists of 103 patients. Of these, 34 (33%) died during hospitalization in the ICU (Figure 1).

By dividing the patients into survivors and the deceased (Table 1), the variables significantly distributed between the two clusters were the LUS score at admission (23.5 vs. 25.7; *p* = 0.02) (Figure 2), days of spontaneous breathing, and days of mechanical ventilation in controlled mode (3.12 vs. 0.24 and 4.84 and 8.47, respectively; *p* < 0.01 for both), age (65 vs. 60, *p* = 0.013) and prevalence of chronic renal failure (*p* = 0.015).

The ROC curve shows an AUC of 61.3% at univariate regression for the LUS score at the admittance (LUSt0). The best cut-off value, according to Youden’s J index method, is 25 (OR 1.1; IQR 1.0−1.2; *p*-value = 0.05; sensitivity 67.6%; specificity 56.5%). 

The stepwise generalized linear regression model shows that the value of LUSt0 (OR 1.1; IQR 1.0−1.3), age (OR 1.1; IQR 1.0−1.2), sex (OR 0.7; IQR 0.2−3.6), and days in spontaneous breathing (OR 0.2; IQR 0.1−0.5) predict the risk of death for COVID-19 patients (AUC 0.92) (Figure 3).

Comparing the LUS score at the two examinations among the survivors, the trend shows a statistically significant reduction in LUS (mean of differences equal to 1.746, *p* = 0.033) (Figure 4).

Further analyzing the data of the LUS among the survivors, we observed that 29 out of 54 patients presented a ΔLUS ([LUSt0-LUSt1]/LUSt0) between −0.10 and −0.39; seven patients showed a ΔLUS below −0.40; 15 patients showed a ΔLUS between −0.09 and 0.09; and finally, three patients showed a ΔLUS greater than 0.10.

## 4. Discussion

The main finding of our study was that COVID-19 patients with a lower LUS score value at admission in the ICU had a better survival rate than the patients who died. 

Analyzing the LUS score trends among the patients that survived, it is possible to identify at least four subpopulations: (a) those whose clinic improved independently from the LUS evolution; (b) those who presented a moderate improvement in the ultrasound imaging; (c) those who responded very clearly, with a significant reduction in pulmonary involvement; (d) those who, while improving their clinical conditions, did not show an evident improvement from an ultrasound point of view and presented an apparent worsening in LUS. The first two subpopulations are the two most represented ones (Figure 1).

Whether this result is due to different disease clusters or the early onset of therapy, it cannot be established from our study design, deserving further targeted studies.

Compared to the literature, the role of the LUS score and, specifically, in COVID-19 pneumonia as a prognostic tool, was investigated in many studies without univocal results. During the COVID-19 pandemic, the spared of LUS use increased. A recent survey conducted on about 700 Italian intensivists, showed that the physicians who use the LUS daily raised from about 10% in the pre-COVID-19 era to 28% in the COVID-19 period. The percentage of daily user intensivists of the LUS score grew from less than 2% to 9%. The majority of the practitioners stated that the LUS influenced their clinical decisions (68%) and patient monitoring (73%) [12]. 

However, the role of the LUS score is still controversial, and conflicting results are present in the literature. Persona et al. did not find in a study that enrolled 28 patients any significant difference in the LUS score at the admission and discharge in survivors and non-survivors, suggesting that the LUS score is not as reliable as in non-COVID-19 ARDS patients [8]. On the contrary, Lithcer et al. found that a higher LUS score predicts mortality and the need for mechanical ventilation [10]. Dargent et al., in a small sample of ten patients with COVID-19 ARDS, showed that the course of the disease could be described with the modifications of the LUS score [13]. Li et al. showed similar results in 280 patients. The LUS score is a useful tool for monitoring patients with COVID-19 ARDS [14]. 

In a study from a Brazilian group, de Alencar et al. found in 180 patients a correlation between the LUS score at admission and death, mechanical ventilation, and intubation. This study considered a broader population spectrum admitted to the emergency department with only 74 ICU patients [15]. In a recent review, a higher baseline LUS score is related to a higher risk of unfavorable outcomes (ICU admission, mechanical ventilation, and death) [16].

Our results agreed with Lithcer, Dargent, Li, and de Alencar, and showed that the LUS score could be used in COVID-19 patients admitted to the ICU as a prognostic tool. Notably, the LUSt0 of 25 is the cut-off value that could predict a higher risk of death. Age and sex are also related to higher mortality risk. Interestingly, more days in spontaneous breathing decrease the mortality risk. 

Our result has a double clinical significance: first of all, it endorses the use of lung ultrasound as a tool to screen COVID-19 patients with respiratory failure to evaluate those most at risk and who therefore require immediate intensive care. Second, monitoring clinical conditions proves to be a useful tool for establishing the effectiveness of ongoing therapies.

However, describing the results obtained, we are aware of the limits of the LUS. Although several authors have described typical patterns of COVID-19 pneumonia [17], to date, it has not yet been shown that the ultrasound picture of COVID-19 patients is particularly different from similar forms of non-COVID-19 interstitial diseases [18]. The specificity of LUS in an audience of general patients is not high [19]. However, the patients entering the ICU are selected patients evaluated by different physicians and imaging techniques. This path could increase the specificity of the LUS [20]. On the other hand, while enjoying higher accuracy, even the chest CT scan shows poor specificity towards SARS-CoV-2 interstitial pneumonia [21].

Furthermore, although beyond the scope of this study, we need to recode the role that LUS has outside the diagnosis or monitoring of COVID-19 patients. LUS is also useful as a guiding tool during invasive procedures: LUS is a particularly suitable tool for bedside procedures, such as the drainage of pleural effusions that complicate the course of COVID-19 patients in a not-so-small percentage [2,22]. This result is particularly relevant if we consider the logistical difficulties of moving a highly infectious patient to the Radiology Department [23,24].

This makes the use of LUS particularly attractive in COVID-19 patients: ultimately, LUS is also an aid for procedures or diagnosis, and allows for prognostic stratification as we suggested with our study.

### Limitations

Our study is retrospective and therefore subject to potential selection bias. Furthermore, only a small percentage of the patients treated and assessed by ultrasound were registered. While confirming the absence of explicit enrollment bias, we cannot exclude any random errors related to the enrollment modality. The study population is not extensive and larger studies are needed to confirm these results.

## 5. Conclusions

Upon entry into the ICU, the lung ultrasound score may play a prognostic role in COVID-19 patients with ARDS. Furthermore, the monitoring employing the LUS score allows for evaluating the patients with a higher probability of survival. A multicentral study is urgently needed to confirm our data.

## Figures and Tables

**Figure 1 healthcare-10-00568-f001:**
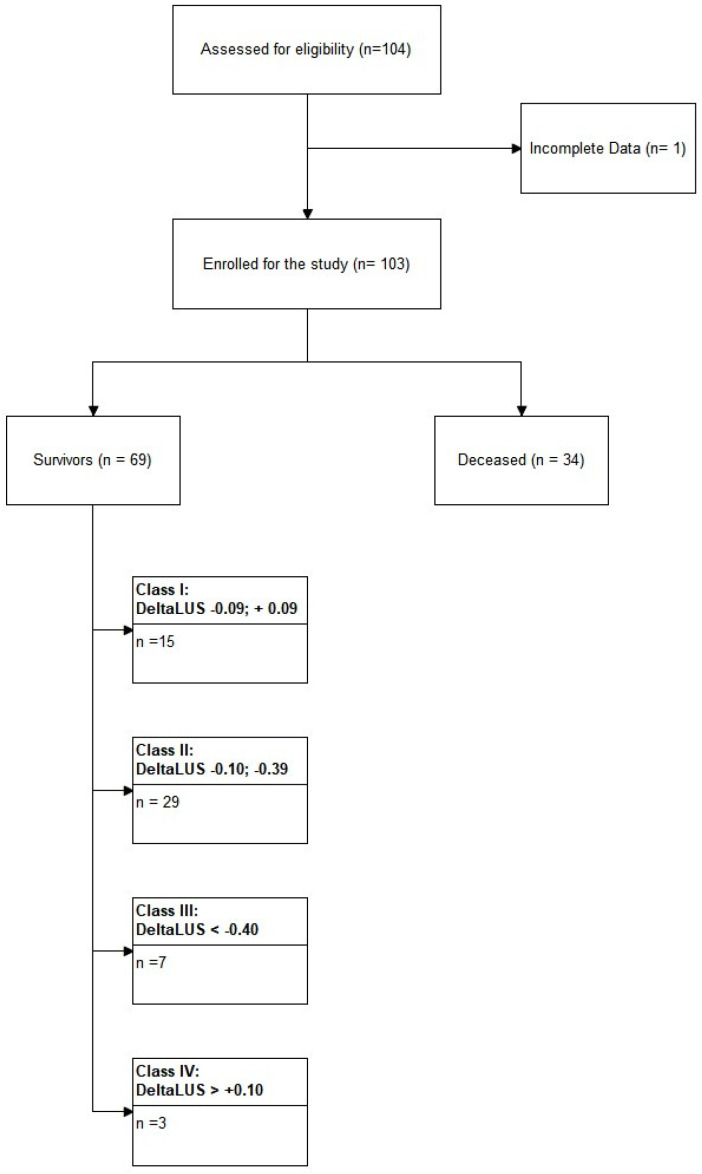
Flowchart of the enrollment process. Surviving patients show an additional 4 classes based on the extent of ΔLUS score magnitude. ΔLUS = delta lung ultrasound.

**Figure 2 healthcare-10-00568-f002:**
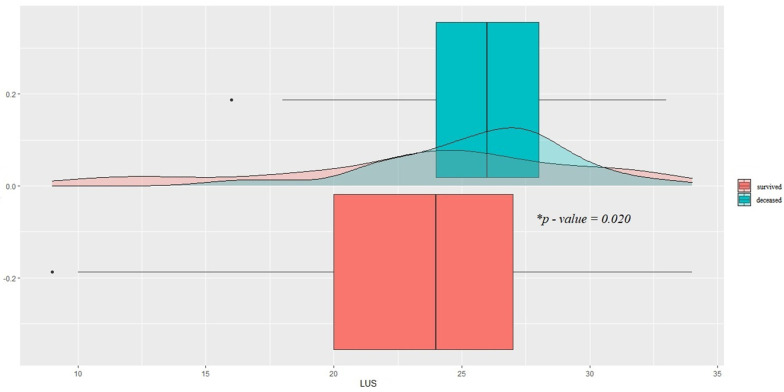
Comparison between LUS score distributions of surviving and deceased patients (23.5 vs. 25.7; * *p* = 0.02). LUS = lung ultrasound.

**Figure 3 healthcare-10-00568-f003:**
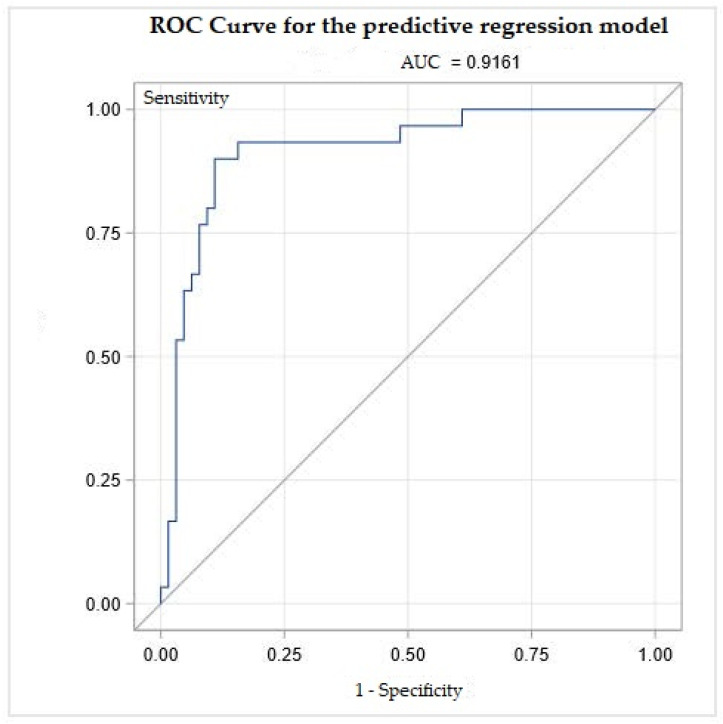
ROC Curve for the predictive regression model. The most predictive variables are LUSt0 (OR 1.1; IQR 1.0−1.3), age (OR 1.1; IQR 1.0−1.2), sex (OR 0.7; IQR 0.2−3.6), and days in spontaneous breath (OR 0.2; IQR 0.1−0.5) predict the risk of death for COVID-19 patients. AUC = Area Under the Curve.

**Figure 4 healthcare-10-00568-f004:**
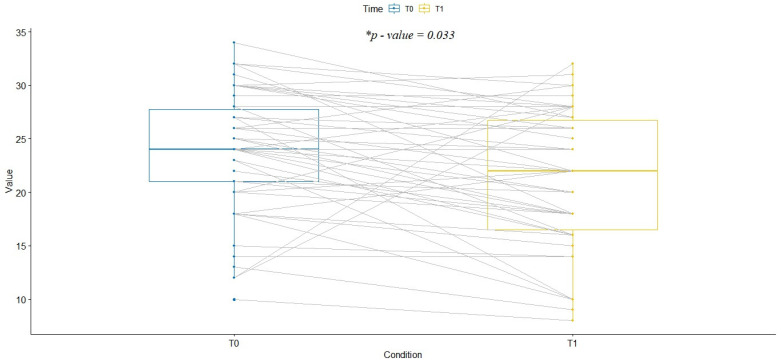
Comparison between LUS score at admission and discharge of surviving patients. The difference is statistically significant (mean difference of 1.75, * *p*-value = 0.03). LUS = lung ultrasound.

**Table 1 healthcare-10-00568-t001:** Clinical, laboratory and anthropometric characteristics of the general population (*n* = 103) and subgroups (surviving patients, *n* = 69; and deceased, *n* = 34) compared by Student’s *t*-test.

	Population	Survived	Deceased	*p*-Value
	*n* = 103	*n* = 69	*n* = 34	
Age (years)	67.0 (10.9)	65.4 (12.0)	70.3 (7.46)	0.013
Sex (male)	81 (78.6%)	54 (78.3%)	27 (79.4%)	NS
BMI	23.0 (14.3)	22.3 (14.4)	24.5 (14.1)	NS
LUS (t0)	24.2 (5.44)	23.5 (6.06)	25.7 (3.54)	0.020
CPR (t0)	11.8 (31.4)	12.7 (35.3)	9.91 (21.8)	NS
Il-6 (t0)	100 (192)	110 (217)	69.6 (62.8)	NS
D-dimer (t0) FEUng/mL	6337 (17,825)	3039 (5958)	13,030 (28,984)	NS
CPR (t1)	6.44 (13.1)	5.97 (15.7)	7.40 (4.97)	NS
Il-6 (t1)	55.2 (77.0)	37.4 (34.3)	111 (133)	NS
D-dimer (t1) FEUng/mL	2310 (3764)	1719 (1966)	3510 (5798)	NS
Spon Breath (days)	2.17 (2.55)	3.12 (2.58)	0.24 (0.85)	<0.001
Controlled Mech. Vent. (days)	6.04 (5.77)	4.84 (6.28)	8.47 (3.54)	<0.001
Assisted Mech. Vent. (days)	3.80 (5.06)	3.97 (5.42)	3.44 (4.30)	NS
Inhaled NO2 (days)	0.91 (3.10)	0.96 (3.57)	0.82 (1.88)	NS
Hypertension	63 (61.2%)	41 (59.4%)	22 (64.7%)	NS
Prev. Myocard. Infarction	5 (4.85%)	3 (4.35%)	2 (5.88%)	NS
Chr. Heart failure	5 (4.85%)	3 (4.35%)	2 (5.88%)	NS
Periph. Vascular disease	11 (10.7%)	6 (8.70%)	5 (14.7%)	NS
Cerebrovascular disease	5 (4.85%)	3 (4.35%)	2 (5.88%)	NS
Cognitive Impairment	1 (0.97%)	1 (1.45%)	0	NS
COPD	17 (16.5%)	9 (13.0%)	8 (23.5%)	NS
Connective tissue diseases	3 (2.91%)	1 (1.45%)	2 (5.88%)	NS
Gastric diseases	1 (0.97%)	0	1 (2.94%)	NS
Mild liver disease	3 (2.91%)	3 (4.35%)	0	NS
Moderate to severe liver disease	1 (0.97%)	1 (1.45%)	0	NS
Diabetes	16 (15.5%)	12 (17.4%)	4 (11.8%)	NS
Diabetes with organ dysfunction	7 (6.80%)	4 (5.80%)	3 (8.82%)	NS
Chronic renal failure	8 (7.77%)	2 (2.90%)	6 (17.6%)	0.015
Solid neoplasm	3 (2.91%)	3 (4.35%)	0	NS
Leukemia	3 (2.91%)	1 (1.45%)	2 (5.88%)	NS
Lymphoma	2 (1.94%)	1 (1.45%)	1 (2.94%)	NS
Autoimmune Diseases	4 (3.88%)	2 (2.90%)	2 (5.88%)	NS
Smoking	14 (13.6%)	9 (13.0%)	5 (14.7%)	NS
Substance Use Disorder	2 (1.94%)	0	2 (5.88%)	NS
Chronic immunosuppressive therapy	6 (5.83%)	3 (4.35%)	3 (8.82%)	NS
Chronic corticosteroid therapy	2 (1.94%)	1 (1.45%)	1 (2.94%)	NS
Home oxygen therapy	3 (2.91%)	3 (4.35%)	0	NS

## Data Availability

Data are available upon request.

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
