# Peer review of "Can Lung Ultrasound Be the Ideal Monitoring Tool to Predict the Clinical Outcome of Mechanically Ventilated COVID-19 Patients? An Observational Study"

_healthcare, 2022, doi:10.3390/healthcare10030568_

Round 1

Reviewer 1 Report

In this work, Ventrugno et al report a small retrospective cohort who had lung ultrasound performed during their hospitalization for severe COVID19. They describe the temporal changes in LUS during hospitalization and confirm my expectation that persistently abnormal lung ultrasound is not associated with healing. While the overall design and conceptualization is sound this work could benefit from either statistical review or focused revisions on the statistical content.  

  • Line 136, Please clarify the type of regression being performed and the resulting coefficients. I suspect this is logistic regression for the event of death. In that case, it would be reasonable to report odd ratios (and confidence intervals) rather than coefficients (in log odds) and standard errors.  
  • For table 1: 
  •  I would suggest clarifying what statistical testing is being performed (I.e., Survived vs. Deceased) and reporting non-significant values as NS rather than explicit p-values.  
  • I would suggest grouping the comorbid conditions into related groups (I.e., place COPD and home oxygen therapy adjacent to each other) and grouping related conditions (I.e, group lymphoma and leukemia, previous MI and CHF) 
  • I would suggest relabeling “Drug Addiction” as “Substance Use Disorder” 
  • I would suggest deleting “Other diseases” 
  • For figure 2: 
  • the axis labels should be in plain English (rather than variable names) 
  •  Y axis is non-sensical (either make it the categorical figure or delete the labels – density is not key to the figure) 
  • Overlapping the density plots with one of the box plots impairs readability 
  • Clarify the statistical test being used – if a T-test is being used then the density plot demonstrates substantial non-normality and a non-parametric test may be more reasonable 

Author Response

          Dear Editor, 

We appreciate the thorough review of our manuscript and the helpful comments provided by the reviewers. You will find a copy of each of the reviewer comments, along with a point-by-point response in bold font. In our reply, we highlighted the line referring to the new manuscript.

Reviewer 1

Comments: In this work, Vetrugno et al report a small retrospective cohort who had lung ultrasound performed during their hospitalization for severe COVID19. They describe the temporal changes in LUS during hospitalization and confirm my expectation that persistently abnormal lung ultrasound is not associated with healing. While the overall design and conceptualization is sound this work could benefit from either statistical review or focused revisions on the statistical content.   Line 136, Please clarify the type of regression being performed and the resulting coefficients. I suspect this is logistic regression for the event of death. In that case, it would be reasonable to report odd ratios (and confidence intervals) rather than coefficients (in log odds) and standard errors.   

Reply: Thank you for this observation. We used a logistic regression and we specified it in the text (Line 138). The type of regression is present as well in the methods section (Line 117).  We have indicated the odds ratios in table 1, as required.

 For table 1:  I would suggest clarifying what statistical testing is being performed (I.e., Survived vs. Deceased) and reporting non-significant values as NS rather than explicit p-values.  I would suggest grouping the comorbid conditions into related groups (I.e., place COPD and home oxygen therapy adjacent to each other) and grouping related conditions (I.e, group lymphoma and leukemia, previous MI and CHF) I would suggest relabeling “Drug Addiction” as “Substance Use Disorder” I would suggest deleting “Other diseases”  

Reply: Following your suggestion, we clarified in Table 1 the caption. We used the Student's T-test (Line 145) after verifying the normality of the distribution by Wilk-Shapiro test (Line 113); and we modified the table according to give more clear and logical readability (Table 1).

 For figure 2: the axis labels should be in plain English (rather than variable names)  Y axis is non-sensical (either make it the categorical figure or delete the labels – density is not key to the figure) Overlapping the density plots with one of the box plots impairs readability Clarify the statistical test being used – if a T-test is being used then the density plot demonstrates substantial non-normality and a non-parametric test may be more reasonable  

Reply: Thank you for this observation. We modified figure 2 according to your note. Unfortunately, our statistical program overlaps the images.

We used T-test and verified the normality of the distribution by means of the Shapiro-Wilk test (Line 113), as mentioned above.

  We would be glad to make further changes upon request. On behalf of the other authors, we extend our gratitude for your time and assistance with our review. 

Kind regards 

Francesco Meroi, MD

Reviewer 2 Report

Journal: Healthcare

Manuscript Number: healthcare-1587571

Title: Can lung ultrasound be the ideal monitoring tool to predict the clinical outcome of mechanically ventilated COVID-19 patients? An observational study.

During the COVID-19 pandemic, using tools to assess disease severity is one of the most important aspects in the emergencies departments. Lung ultrasound (LUS) is widely used because it is user-friendly, broadly available, low-cost, and has a high accuracy for diagnosing pulmonary diseases and also for monitoring its severity and the effects of maneuvers and therapies.

Comments

  • The present study is of great interest since there are only few studies demonstrating the useful of LUS in predicting outcomes in patients with COVID-19. Particularly, the LUS score is a semiquantitative score that measures lung aeration loss caused by different pathological conditions, usually used to quantify lung involvement; in this article, authors demonstrated its strong correlation with mortality in the peculiar setting of COVID-19 pandemic. However, study population of 103 patients is not so broad as to allow for generalization; this aspect should be mentioned in study limitations.
  • Interestingly, authors demonstrated that the deceased patients had a higher LUS score at admission than the survivors (25.7 vs 23.5). It should be interesting to correlate these results to a more “objective” data from a chest CT-scan, considering that this one is the gold standard for diagnosing COVID-19 pneumonia. Indeed, it should be very important if LUS correlated with the estimated extent of parenchymal involvement found from chest CT, assuming that higher mortality correlates with greater extension of pneumonia. Can authors add these data?
  • The primary endpoint of the study was mortality. Can authors add some secondary endpoints such as endotracheal intubation for respiratory failure?
  • Similar previous studies have recently published (for example: Julio Cesar Garcia de Alencar et al, Lung ultrasound score predicts outcomes in COVID-19 patients admitted to the emergency department; Annals of Intensive Care; 2021) with similar results; please expand the discussion with more recent (in the last years) published studies.
  • A "wet lung" detected by lung ultrasound by several B-lines also predicts impending acute heart failure decompensation; can authors add some data about the “cardiac” history of enrolled patients? It should be interesting to discuss a possible cardiac components that could confound the obtained results.
  • In the manuscript there are some grammar errors and some sentences are quite confounding with a poor English language. Please briefly review the language of entire article.

Author Response

Dear Editor, We appreciate the thorough review of our manuscript and the helpful comments provided by the reviewers. You will find a copy of each of the reviewer comments, along with a point-by-point response in bold font. In our reply, we highlighted the line referring to the new manuscript.

  Reviewer 2   The present study is of great interest since there are only few studies demonstrating the useful of LUS in predicting outcomes in patients with COVID-19. Particularly, the LUS score is a semiquantitative score that measures lung aeration loss caused by different pathological conditions, usually used to quantify lung involvement; in this article, authors demonstrated its strong correlation with mortality in the peculiar setting of COVID-19 pandemic. However, study population of 103 patients is not so broad as to allow for generalization; this aspect should be mentioned in study limitations.  

Reply: Thank you for this observation. We added this sentence “The study population was not extensive and larger studies are required to confirm these results.” (Line 229) in the study limitations.  

Interestingly, authors demonstrated that the deceased patients had a higher LUS score at admission than the survivors (25.7 vs 23.5). It should be interesting to correlate these results to a more “objective” data from a chest CT-scan, considering that this one is the gold standard for diagnosing COVID-19 pneumonia. Indeed, it should be very important if LUS correlated with the estimated extent of parenchymal involvement found from chest CT, assuming that higher mortality correlates with greater extension of pneumonia. Can authors add these data?  

Reply: We thank the reviewer for this advice. Many studies demonstrated the correlation between CT scans and LUS scores. Instead, we designed our study focusing the attention on the lung ultrasound score used to reduce CT scan and ionizing radiation. We did not record data on the CT scan. The specific correlation between the two techniques is well documented in the literature, both in the "classic" patient and in the COVID-19 patient. We aimed to determine whether the LUS score could play a role in reducing CT scans and predicting a COVID-19 patient's prognosis.   

The primary endpoint of the study was mortality. Can authors add some secondary endpoints such as endotracheal intubation for respiratory failure?  

Reply: We apologize to the reviewer; due to the study's retrospective nature, these parameters were not present in the excel file. This does not mean that this outcome will be duly taken into consideration in the next analyses that we could carry out.  

Similar previous studies have recently published (for example: Julio Cesar Garcia de Alencar et al, Lung ultrasound score predicts outcomes in COVID-19 patients admitted to the emergency department; Annals of Intensive Care; 2021) with similar results; please expand the discussion with more recent (in the last years) published studies.  

Reply: Following your suggestion, we expanded the discussion including the reference above. (Line 193)  

A "wet lung" detected by lung ultrasound by several B-lines also predicts impending acute heart failure decompensation; can authors add some data about the “cardiac” history of enrolled patients? It should be interesting to discuss a possible cardiac components that could confound the obtained results.  

Reply: Five patients in our study population had a history of chronic heart failure, and five had a history of myocardial infarction, as shown in Table 1. The lung involvement of the cardiac disease is different from ARDS. The cardiac “wet lung” is homogeneous from the apex to the base in all regions, while ARDS shows typical spared areas and peculiar signs. (Copetti R, Soldati G, Copetti P. Chest sonography: a useful tool to differentiate acute cardiogenic pulmonary edema from acute respiratory distress syndrome. Cardiovasc Ultrasound. 2008 Apr 29;6:16. doi: 10.1186/1476-7120-6-16.) Furthermore, given the smallness of the sample, linked to the fact that ours is an exploratory analysis, we are unfortunately not able to verify the particular susceptibility of some sub-class with a certain margin of statistical reliability.  

In the manuscript there are some grammar errors and some sentences are quite confounding with a poor English language. Please briefly review the language of entire article.     

Reply: As suggested by the reviewer, we revised once more the grammar and the English language of the entire article.   We would be glad to make further changes upon request. On behalf of the other authors, we extend our gratitude for your time and assistance with our review.   Kind regards   Francesco Meroi, MD

Round 2

Reviewer 1 Report

The authors have responded to the comments and made appropriate revisions. 

Author Response

          Dear Editor,

We thank you for the review of our manuscript and the helpful comments. Following your suggestions, we asked Dr. Valent of the University of Trento, very valid biostatistics, to review the entire statistical analysis of our paper. For the substantial contribution of Dr. Valent, we ask to include her in the list of authors. All authors agree to include Dr. Valent for her important contribution and critical revision of the manuscript. We provide as well her statistical analysis in the following files.

Editor:

In this work, Vetrugno et al describe retrospectively a cohort of patients hospitalized for severe COVID-19 and focused on the utility of lung ultrasound performed at the admission and discharge from ICU to predict deterioration (death). The matter is particularly interesting, considering the wide spread of use of lung ultrasound (COVID-19 pandemic has contributed) to easily assess lung involvement in multiple diseases. As, Reviewer 1 has pointed out, main of the concerns are about design of the study (retrospective) and statistical analysis. Particularly this latter is the weak point ant still it remains after revision by authors according to reviewer suggestions. Given that statistics is not my main topic of research and reading introduction and aims explicated by authors in introduction and aims of the study, I expected a different statistical approach and results. If the main aim of the study was to assess validity of LUS (I mean initial LUS) to predict the outcome, I would expect some result that might help the clinician to forecast the outcome of the patient (at least probabilistically). In other words can the author state that a “determinate” cut-off value of the variable clearly, measured at the initial assessment, discriminate if a patient will fall in which of the two groups? I think that a logistic regression model is likely to be the most appropriate, as already pointed out by the reviewer 1. The answers to the questions of reviewer 1 are not fully convincing. I feel that, at last, just a simple t-test (post-pre value of lung score) is supporting the conclusions of the authors. From a very practical point of view, let’s suppose to have a patient with an initial lung score of 25, would the authors consider the patient at a greater risk of dying than if he had 25? Moreover we have only marginally considered that inter-rater variability of Lung Ultraosund greatly depends on experience of the performer (I think we can agree that 1-2 points of difference in LUS cannot be reliable to answer such a difficult question). Just to give a practical sense to my final revision, I would suggest the authors to consider (if not already done with the help o f a statistician) a real logistic regression model in which multiple parameters chosen on the basis of a discriminant analysis can produce a more accurate model. If not possible (and this is likely if not done until now) conclusions of the authors should be more and more cautious.

Reply: Dr. Valent performed univariate regression again to identify the best cut-off value for the LUS, which was 25. In addition, it applied a stepwise logistics regression model to identify predictive variables. Again, the LUS was a predictive variable. In addition, the obtained model consisting of the viable LUS, age, sex and days in spontaneous respiration obtained an AUC of 92%.

We implemented the paper with the results (Line 171) obtained and added the new Figure 3 (Line 179), which shows the predictive model's ROC curve obtained by multivariate analysis. We labeled the previous figure 3 as figure 4 (Line 189).

According to the results described, we briefly implemented the discussion (Line 243).

   We would be glad to make further changes upon request.

On behalf of the other authors, we extend our gratitude for your precious suggestions and assistance with our review.

   Kind regards

   Francesco Meroi, MD
